# Citroflavonoids as Promising Agents for Drug Discovery in Diabetes and Hypertension: A Systematic Review of Experimental Studies

**DOI:** 10.3390/molecules27227933

**Published:** 2022-11-16

**Authors:** Rolffy Ortiz-Andrade, Jesús Alfredo Araujo León, Juan Carlos Sánchez-Salgado, Amanda Sánchez-Recillas, Priscila Vazquez-Garcia, Emanuel Hernández-Núñez

**Affiliations:** 1Laboratorio de Farmacología, Facultad de Química, Universidad Autónoma de Yucatán, Merida 97069, Mexico; 2Unidad de Bioquímica y Biología Molecular de Plantas, Centro de Investigación Científica de Yucatán, A.C., Calle 43 No. 130 x 32 y 34, Col. Chuburná de Hidalgo, Mérida 97205, Mexico; 3Hypermedic MX, Ciudad de Mexico 04930, Mexico; 4Departamento de Recursos del Mar, Centro de Investigación y de Estudios Avanzados del Instituto Politécnico Nacional-Unidad Mérida, Merida 97205, Mexico

**Keywords:** biochemical parameters, blood markers, gene expression, hesperidin, naringenin, protein content

## Abstract

Flavonoids are naturally occurring compounds widely distributed in the *Citrus* genus. These natural compounds have many health benefits, mainly for metabolic and cardiovascular diseases. In fact, some these compounds are components of drug products with approved indications for peripheral vascular insufficiency and hemorrhoids. However, information on pharmacological effects of these compounds remains disperse and there is scarce comprehensive analysis of whole data and evidence. These kinds of evidence analyses could be necessary in drug design and the development of novel and innovate drug products in diabetes and hypertension. We aimed to systematically search for evidence on the efficacy of citroflavonoids in diabetes and hypertension in in vivo models. We searched four literature databases based on a PICO strategy. After database curation, twenty-nine articles were retrieved to analyze experimental data. There was high heterogeneity in both outcomes and methodology. Naringenin and hesperetin derivates were the most studied citroflavonoids in both experimental models. More investigation is still needed to determine its potential for drug design and development.

## 1. Introduction

Flavonoids comprise one of the significant natural compounds that have been interesting for researchers in the last two decades. They are a class of secondary metabolites widely spread in nature and found in several fruits and vegetables, mainly in Citrus genus [1]. Citroflavonoids (flavonoids extracted from *Citrus* species) are molecules characterized by multiple health benefits and biological activities, such as anticarcinogenic, antihyperglycemic, antihyperlipidemic, neuroprotective, and hepatoprotective activities reported in experimental and clinical frameworks. Hesperidin, nobiletin, naringin, diosmin, quercetin, rutin, hesperetin, naringenin and tangeretin are major citroflavonoids identified in nature [1]. It is common to find dietary flavonoids used as glycosylated derivates. The loss of glycosyl moiety is a critical step in the absorption and metabolism of citroflavonoids [2]. In fact, only aglycones and some glycosides can be absorbed. This evidence shows the importance of first-step metabolism on citroflavonoid bioactivity [3].

Some systematic reviews have exposed the impact of citroflavonoid consumption on cardiovascular disease and diabetes. [4,5,6]. An extensive systematic review found a reverse association between the consumption of flavonoids and a reduction in the risk of cardiovascular disease [6]. Until now, no published systematic review has assessed the effect of flavonoid-rich fruits on blood pressure [7]. On the other hand, although the association between the consumption of citrus fruits and the incidence of diabetes is not fully understood, some systematic reviews have shown that this behavior has a protective role [8].

Interestingly, some nutraceuticals have been developed for treating prediabetes as therapeutic alternatives. Eriomin^®^, a formula of citrus flavonoids composed of three single flavonoids, is a marketed end-product developed in Brazil. A double-blind, randomized, controlled trial showed evidence that this flavonoid mixture produces anti-inflammatory, antihyperglycemic, and antioxidant in those patients [9]. Furthermore, diosmin and hesperidin are well known components of the approved medicine Daflon^®^, a medication indicated for the treatment of peripheral vascular insufficiency and hemorrhoids [10]. An example such as this should be used as an encouraging factor to promote drug development of new and improved flavonoid-based end-products achievable for the population with cardiovascular risk factors.

Recently, systematic review and meta-analysis have become increasingly important in decision-making in the research field [11]. However, information on pharmacological effects of citroflavonoids remains disperse and there is scare comprehensive analysis of whole data and evidence. These kinds of evidence analyses could be necessary in drug design and the development of novel and innovate drug products in diabetes and hypertension. This work compares the efficacy outcomes reported for citroflavonoids evaluated in experimental diabetes and hypertension animal models. This investigation could be used as a reference for deciding what flavonoid is most promising for drug development and learning more about state-of-the-art flavonoid research.

## 2. Results

Overall, 468 articles were identified in primary searching (Pubmed = 110, Scopus = 187, Web of Science = 161, Lilacs = 10). After duplicate elimination, 283 unique articles remained for an abstract, title, and full-text revision. Only 29 articles (21 for diabetes and 8 for hypertension) remained for data analysis and discussion (Figure 1).

After full-text revision of selected articles, we found that the most studied flavonoids in diabetes and hypertension animal models were hesperidin (diabetes: 7, hypertension: 4), naringin (diabetes: 6, hypertension: 0), naringenin (diabetes: 5, hypertension: 1), diosmin (diabetes: 2, hypertension: 1), nobiletin (diabetes: 1, hypertension: 2), glycosyl-hesperidin (diabetes: 0, hypertension: 2), hesperitin (diabetes: 1, hypertension: 1), targeretin (diabetes: 1, hypertension: 0), sudachitin (diabetes: 1, hypertension: 0), apigenin (diabetes: 0, hypertension: 1), and neohesperidin (diabetes: 1, hypertension: 0). A list of these molecules is positioned below (Figure 2).

Hesperidin, naringin, and naringenin (naringin aglycone) were the most studied citroflavonoids. Plasma biomarkers of carbohydrate and lipid metabolism, such as glucose and lipidic molecules (cholesterol, triglycerides, lipoproteins), liver function biomarkers (TGP, TGO, and alkaline phosphatase), were the most evaluated parameters for determining the bioactivity of citroflavonoids in diabetes. Moreover, body weight changes over time were assessed in almost all studies despite the low activity on this parameter. In addition, functional metabolic assays were also performed to establish changes in glucose uptake, insulin tolerance, and other metabolic pathways (Table 1).

**Table 1 molecules-27-07933-t001:** Study design of selected articles on diabetes animal models.

No	Compound Name	Dosing	Follow Up Period	Animal Model		Study Outcomes	Reference
				Type of Animal	Age/Weight		
1	Naringin	100 mg/kg	4 weeks	STZ-NA Wistar rat	130–150 g	Blood biomarkers, OGTT, liver-specific enzyme activity, gene expression, and biomarkers content	[12]
	Naringenin						
2	Hesperidin	10 mg/kg (1%) *	4 weeks	Goto-Kakizaki rat	3 weeks old	Blood biomarkers, liver-specific enzyme activity and gene expression	[13]
3	Hesperidin	10 g/kg (1%) *	4 weeks	STZ Wistar rat	3 weeks old	Blood biomarkers, tissue-specific biomarkers content, and enzyme activity, serum flavonoid content, bone density	[14]
4	Naringenin	30 mg/kg (3%) *	4 weeks	Lep ob/ob mouse	8–12 weeks old	Blood biomarkers, OGTT, ITT, tissue-specific biomarkers content, and gene expression, IHC analysis, anthropometric micro-CT imaging	[15]
5	Naringenin	30 mg/kg (3%) *	12 weeks	C57BL/6J Ldlr^−/−^ mouse	10–12 weeks old	Blood biomarkers, tissue-specific biomarkers content and gene expression, OGTT, ITT, IHC analysis, flow cytometry analysis for cell identification	[16]
6	Hesperetin	40 mg/kg	6 weeks	Wistar rat	170–200 g	Blood biomarkers, tissue-specific biomarkers content, IHC analysis, oxidative stress assessment	[17]
7	Neohesperidin	50 mg/kg	6 weeks	KK-Ay mice	8 weeks old	OGTT, ITT, blood biomarkers, liver-specific parameters, tissue-specific gene expression and protein quantification	[8]
8	Hesperidin	0.2 g/kg *	5 weeks	db/db mice	5 weeks old	Blood biomarkers, liver-specific parameters, IHC analysis	[18]
	Naringin				(23 g)		
9	Hesperidin	0.2 g/kg *	5 weeks	db/db mice	5 weeks old	Blood biomarkers, tissue-specific gene expression, protein quantification, and enzyme activity	[19]
	Naringin						
10	Hesperidin	50 mg/kg	4 weeks	HFD/STZ-Wistar rat	180–200 g	OGTT, blood biomarkers, liver-specific enzyme activity, in situ intestinal glucose absorption, in vitro insulin secretion test	[20]
	Naringin						
11	Naringin	100 mg/kg	4 weeks	High fructose diet Sprague- Dawley rat	180–200 g	Vascular reactivity, blood biomarkers, artery-specific protein quantification	[21]
12	Naringenin	10, 30 mg/kg (1% or 3%) *	4 weeks	C57BL/6J Ldlr^−/−^ mouse	12 weeks old	OGTT, ITT, blood biomarkers, tissue-specific parameters, gene expression, and enzyme activity, in situ intestinal lipid absorption, energy expenditure	[22]
13	Nobiletin	10, 30 mg/kg (1% or 3%) *	8–26 weeks	C57BL/6J Ldlr^−/−^ mouse	12 weeks old	Tissue- and cell-specific gene expression, blood biomarkers, hyperinsulinemic—euglycemic clamp and pyruvate tolerance test energy expenditure	[23]
14	Naringenin	6, 12.5, 25 mg/kg	6 weeks	HFD/STZ Wistar rat	12–13 weeks old (75–100 g)	Blood biomarkers, lipid peroxidation determination, tissue-specific enzyme activity, gene expression, and protein quantification, IHC analysis	[24]
15	Naringin	50, 100, 200 mg/kg *	3 weeks	HFD/STZ Wistar rat	150–200 g	Blood biomarkers, tissue-specific lipid determination, enzyme activity, and gene expression	[25]
16	Diosmin	100 mg/kg	6 weeks	STZ Wistar rat	200–220 g	Blood biomarkers, tissue-specific enzyme activity, IHC analysis	[26]
17	Diosmin	100 mg/kg	6 weeks	STZ Wistar rat	180–220 g	Blood biomarkers, tissue-specific lipid determination and enzyme activity	[27]
18	Targeretin	100 mg/kg	4 weeks	STZ Wistar rat	180–200 g	OGTT, blood biomarkers, liver-specific enzyme activity and glycogen content, IHC analysis	[28]
19	Hesperidin	25, 50, 100 mg/kg	4 weeks	STZ Wistar rat	200–220 g	OGTT, blood biomarkers, liver-specific enzyme activity and glycogen content, IHC analysis	[29]
20	Hesperidin	20 ppm **	2 weeks, 3 days	STZ albino mouse †	~30 g	Blood biomarkers, malformations rate, number of diabetic fetuses, whole body staining analysis	[30]
21	Sudachitin	5 mg/kg	12 weeks	HFD and db/db mouse	4 weeks old	Blood biomarkers, anthropometric computed tomography analysis, OGTT, ITT, tissue-specific gene expression, energy expenditure	[31]

Note: On some studies flavonoid preparations were supplemented by pellet-based diet (*) or drinking water (**). † This study was performed on pregnant diabetic mice. HFD: high-fat diet; IHC: immunohistochemistry staining technique; ITT: insulin tolerance test; OGTT: oral glucose tolerance test, STZ: streptozotocin-induced diabetic animal, STZ-NA: streptozotocin-nicotinamide-induced diabetic animal.

Conversely, tail blood pressure measurement was the significant endpoint studied in hypertensive animal models. In addition, vascular reactivity to contraction or relaxation inducers in vessels extracted from these animals was the second most evaluated endpoint (Table 2). A comprehensive and detailed set of experimental data from each selected article is described in Appendix A.

**Table 2 molecules-27-07933-t002:** Study design of selected articles on hypertension animal models.

No	Compound Name	Dosing	Treatment Duration	Animal Model		Efficacy Outcome	Reference
				Type of Animal	Age/Weight		
1	Hesperidin	50 mg/kg	4 weeks	SHR rat	15 weeks	Blood pressure changes, vascular reactivity	[32]
2	Nobiletin	20, 40 mg/kg *	4 weeks	SHR (stroke prone) rat	7 weeks	Blood pressure changes, cerebral vessels thrombogenesis test	[33]
3	Hesperidin	30 mg/kg *	25 weeks	SHR rat	3 weeks	Blood pressure and heart rate changes	[34]
	Glucosyl hesperidin						
4	Apigenin	1.44 mg/kg **	6 weeks	L-NAME Sprague–Dawley	300–325 g	Blood pressure and heart rate changes, vascular reactivity, IHC analysis	[35]
	Diosmin	7.16 mg/kg **					
5	Nobiletin	20, 40 mg/kg	2 weeks	L-NAME Sprague–Dawley	220–250 g	Conscious and unconscious blood pressure changes, vascular reactivity, artery-specific protein quantification, blood nitrate/nitrite quantification, IHC analysis	[36]
6	Hesperidin-naringenin mixture	150 mg/kg	4 weeks	SHR rat	250–300 g	Blood pressure and heart rate changes, vascular reactivity	[37]
7	Hesperidin	1 mg/kg *	12 weeks	HFaD Apo-E KO mouse	9 weeks old	Blood biomarkers, vascular reactivity, IHC analysis	[38]
	Glucosyl hesperidin	5 mg/kg *					
8	Hesperidin	20, 40 mg/kg	4 weeks	One-clipped kidney Sprague-Dawley rat	150–180 g	Conscious and unconscious blood pressure changes, blood biomarkers, vascular reactivity, artery-specific enzyme activity and protein content, blood nitrate/nitrite quantification	[39]

Note: On some studies flavonoid preparations were supplemented by pellet-based diet (*) or drinking water (**). This study was performed on pregnant diabetic mice. HFD: high-fat diet; IHC: immunohistochemistry staining technique; ITT: insulin tolerance test; OGTT: oral glucose tolerance test, STZ: streptozotocin-induced diabetic animal, STZ-NA: streptozotocin-nicotinamide-induced diabetic animal.

## 3. Discussion

Citrus fruits are rich in flavonoid compounds, such as hesperidin, hesperetin, naringin, naringenin, diosmin, quercetin, rutin, nobiletin, tangeretin, and others. Mainly, citroflavonoids are present in many citrus fruits, such as bergamots, grapefruit, lemons, limes, mandarins, oranges, and pomelos [20]. Since the early 1950s, flavonoid research has been continuously growing due to the widespread health benefits found for these natural compounds and their applicability in managing non-communicable and infective diseases. Interestingly, from 1996 to the present, the investigation of these natural compounds has increased exponentially (Appendix A). However, evidence searching based on systematic reviews has arisen in the last two decades (Appendix A). For this reason, our systematic review focused on updating and acquiring more knowledge about the biological effects reported in experimental studies extensively.

Our systematic review showed that hesperidin or its aglycone was used in doses from 10 to 100 mg/kg, as naringin and its aglycone were administered. There was not a reasonable explanation as to why researchers selected this dose range. However, regarding other systematic reviews on flavonoid bioactivity, this dose range is employed for any experimental study that evaluates bioactivity in animal species [29].

Several studies investigate the effect of flavonoid-rich fruit extract on blood pressure but with conflicting results. Moreover, herbal extracts are composed of multiple compound groups, mostly polyphenol-like structures that have been to prove also improve cardiovascular and metabolic status in patients [32].

Other findings showed that STZ-induced diabetic rat (Sprague Dawley or Wistar strain) and SHR models were the most used for flavonoid bioactivity assessment. Notably, although the Sprague Dawley model was developed from the Wistar strain, data show that the Wistar strain is prone to metabolic impairments [33]. For this reason, a Sprague Dawley model might be a more acceptable and similar model of diabetes than those on Wistar rats.

The analysis showed a variety of oral administration (by drinking water, enriched diet, or intragastric administration) and time framework (4–25 weeks). Furthermore, using different vehicles such as distilled water and carboxymethylcellulose might be a bias factor. For this reason, a meta-analysis was not possible. Additionally, animal species were possibly crucial for representing biological behavior and processes than chemical- or diet-induced models to surgery-based models with unreal conditions of essential hypertension (renal involved hypertension) [31]. The most used animal models for diabetes and hypertension were streptozotocin (STZ)-induced diabetic rat and spontaneously hypertensive rat (SHR) models.

Currently, some articles have systematically searched to analyze natural products’ proficiency in specific illnesses (e.g., triterpenes on wound healing). Recently, Chen et al. reviewed phenylated flavonoids and their biological effects; this investigation focused on the structural properties of the phenylated flavonoids and the implication for biological activity [17].

Finally, the precise mechanism by which flavonoids exert their antidiabetic and/or antihypertensive effects is unclear. However, evidence suggests that flavonoids might improve the oxidant/antioxidant status imbalance. For example, Jayaraman et al. reported that hesperetin might improve the antioxidant capacity of the liver tissue by restoring enzymatic activity and content [22]. On the other hand, it has been reported that citroflavonoids regulate inflammatory response and deposition of extracellular matrix [12]. More investigation on mechanistic evaluations must be encouraged.

## 4. Materials and Methods

### 4.1. Literature Sources

We performed a systematic search of the literature in four major databases: PubMed (Medline), Scopus, Web of Science, and Lilacs. PRISMA statement was followed to establish the evidence-based minimum set of items for reporting on systematic review (Figure 3).

### 4.2. Eligibility Criteria

A PICO strategy was developed as described in Appendix A. Briefly, the databases were searched using combinations of the following terms: citrus, flavonoid, citroflavonoid, diabetes, and hypertension. A filter was applied to retrieve only experimental animal-based studies up to 2019. Clinical trials, case reports, narrative reviews, editorial letters, and comments were excluded. Systematic reviews and meta-analyses were used only for comparative analysis and results in discussion. Duplicated articles and those without abstract or full-text documents were also excluded from this study. Language limits were not considered.

### 4.3. Studies Selection

The inclusion criteria used for selecting studies are described below. All articles containing the above terms in title, abstract or full text and those reporting experimental outcomes on diabetes and hypertension induced by mixed or single citroflavonoids were selected. Only studies where oral administration is employed were included. The selection was conducted by two independent reviewers for discrepancies.

### 4.4. Meta-Analysis

A meta-analysis of the retrieved data could not be carried out due to methodological heterogeneity such as dosing, mode of administration, animal species, and exposure time.

### 4.5. Risk of Bias Assessment

To assess the risk of biases in the selected studies, we applied a modified version of Cochrane’s RoB tool, the so-called Systematic Review Centre for Laboratory animal Experimentation (SYRCLE). This tool considers six biases: selection, performance, detection, attrition, reporting, and other biases [12,40].

## 5. Conclusions

Systematic searching and comprehensive literature analysis should be the first-line task in pharmacology research to perform an evidence-based decision and to know what compound or substance family needs to be investigated and what will be the way and strategy to be applied to produce more original and relevant data. Furthermore, systematic analysis of evidence could be a promising area to find well-known lead molecules able to be repositioned in the clinical management of hypertension and diabetes.

## Figures and Tables

**Figure 1 molecules-27-07933-f001:**
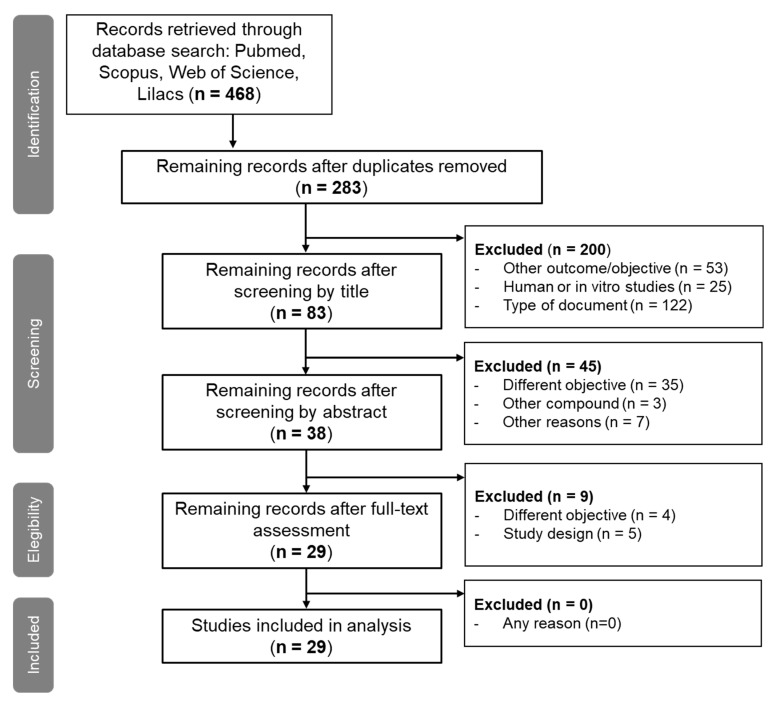
PRISMA flowchart of the selection process of articles that fulfilled the criteria.

**Figure 2 molecules-27-07933-f002:**
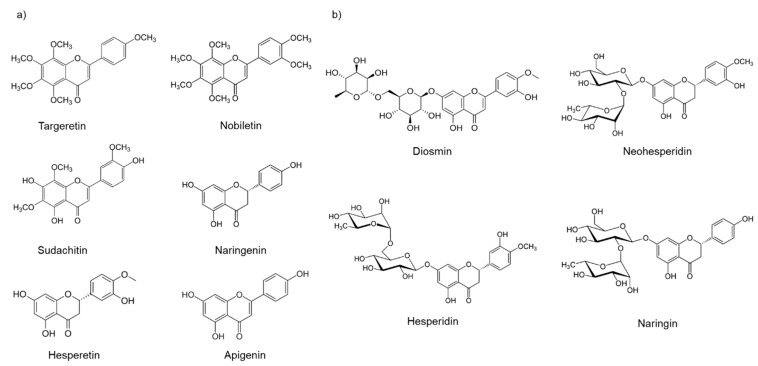
Chemical structures of the citroflavonoids were found in selected articles of the systematic searching; (**a**) aglycones, (**b**) glycosides. Sketched structures were designed by ChemDraw Professional 20 software (PerkinElmer, Inc., Waltham, MA, USA).

**Figure 3 molecules-27-07933-f003:**
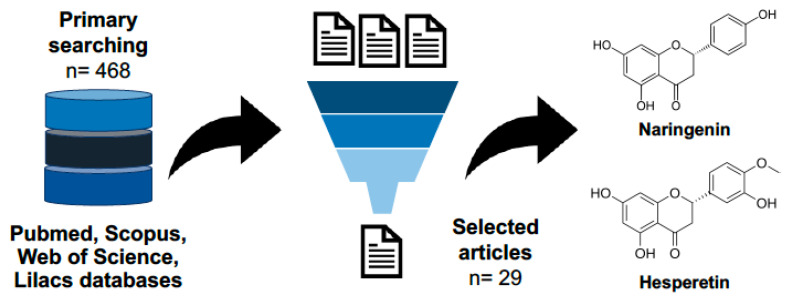
Graphic representation of the methodological process applied to perform this systematic review.

## Data Availability

Not applicable.

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
