# Peer review of "Citroflavonoids as Promising Agents for Drug Discovery in Diabetes and Hypertension: A Systematic Review of Experimental Studies"

_molecules, 2022, doi:10.3390/molecules27227933_

Round 1
Reviewer 1 Report
I have gone through the manuscript ‘Citroflavonoids as promising agents for drug discovery in diabetes and hypertension: An experimental studies-based systematic review’. Overall, the article is well written. I have a few suggestions to improve it to match the expectations of the Journal.
1. The abstract should somewhere highlight the importance or novelty of this review.
2. Authors are suggested to improve the Introduction. Focus more on- What are the gaps in knowledge, need of your systemic review, its novelty and importance.
3. The discussion section should be strengthened by providing remarks on the strengths and weaknesses of reported articles covered in this systematic analysis.
4. In Figure 2, which tools are used to prepare chemical structures? Mention it in legend or text.
5. Based on information gathered through this systematic analysis, what future perspectives the authors put forward. The conclusion section may also include few future avenues or leads that we get from the current report.
6. Spacing, punctuation marks, grammar, and spelling errors should be reviewed thoroughly. I found so many typos throughout the manuscript.
Author Response
Reviewer 1
I have gone through the manuscript ‘Citroflavonoids as promising agents for drug discovery in diabetes and hypertension: An experimental studies-based systematic review’. Overall, the article is well written. I have a few suggestions to improve it to match the expectations of the Journal.
- The abstract should somewhere highlight the importance or novelty of this review.
Answer: Thank you for your comments. We have improved this section based on your suggestions.
- Authors are suggested to improve the Introduction. Focus more on- What are the gaps in knowledge, need of your systemic review, its novelty and importance.
Answer: Thank you for your comments. We have improved this section based on your suggestions.
- The discussion section should be strengthened by providing remarks on the strengths and weaknesses of reported articles covered in this systematic analysis.
Answer: Thank you for your proposal. However, strengths and weaknesses were considered in a global manner for entire selected articles.
- In Figure 2, which tools are used to prepare chemical structures? Mention it in legend or text.
Answer: Thank you for your comments. We have included a text describing this requirement.
- Based on information gathered through this systematic analysis, what future perspectives the authors put forward. The conclusion section may also include few future avenues or leads that we get from the current report.
Answer: Thank you for your comments. We have improved this section based on your suggestions.
- Spacing, punctuation marks, grammar, and spelling errors should be reviewed thoroughly. I found so many typos throughout the manuscript.
Answer: Thank you for your comments. We have checked errors in all manuscript body.
Reviewer 2 Report
The article “Citroflavonoids as promising agents for drug discovery in diabetes and hypertension: An experimental studies-based systematic review”, is an interesting comprehensive systematic review. Presentation through pictures and tables enabled an easy and comprehensive overview of the work. I have found a general good quality of the research, well organized and clearly described. In my opinion, the quality of this manuscript is acceptable to be published in a Molecules journal, after minor revision:
Section Conclusions need revision and some additional sentences, to emphasize findings about health benefits of citroflavonoids, their influence on diabetes and hypertension. Conclusion should be connected with the title and abstract. In order to complement the paper and further improve it, I think the authors should insert a section (tables) related to extraction and analytical techniques for the determination of these group of flavonoids.
Author Response
Reviewer 2
The article “Citroflavonoids as promising agents for drug discovery in diabetes and hypertension: An experimental studies-based systematic review”, is an interesting comprehensive systematic review. Presentation through pictures and tables enabled an easy and comprehensive overview of the work. I have found a general good quality of the research, well organized and clearly described. In my opinion, the quality of this manuscript is acceptable to be published in a Molecules journal, after minor revision:
- Section Conclusions need revision and some additional sentences, to emphasize findings about health benefits of citroflavonoids, their influence on diabetes and hypertension. Conclusion should be connected with the title and abstract. In order to complement the paper and further improve it.
Answer: Thank you for your comments. We have improved both abstract and conclusion sections.
- I think the authors should insert a section (tables) related to extraction and analytical techniques for the determination of these group of flavonoids.
Answer: Thank you for your suggestions. Actually, we are writing a new systematic review aimed to describe all analytical techniques (HPLC, MS, etc) used to quantify citroflavonoids and their comparisons between them.